# Digital contact tracing and network theory to stop the spread of COVID-19 using big-data on human mobility geolocalization

**Matteo Serafino**[1,2], **Higor S. Monteiro**[3], **Shaojun Luo**[1], **Saulo D. S. Reis**[3],
**Carles Igual**[4], **Antonio S. Lima Neto**[5,6], **Matías Travizano**[7], **José S. Andrade Jr**[3],
**Hernán A. Makse**[1] *

**1** Levich Institute and Physics Department, City College of New York, New York, New York, United States of America, **2** IMT School for Advanced Studies, Lucca, Italy, **3** Departamento de Física, Universidade Federal do Ceará, Fortaleza, Brazil, **4** Instituto de Telecomunicaciones y Aplicaciones Multimedia (ITEAM), Departamento de Comunicaciones, Universitat Politècnica de València, València, Spain, **5** Secretaria Municipal de Saúde de Fortaleza (SMS-Fortaleza), Fortaleza, Brazil, **6** Centro de Ciências da Saúde, Universidade de Fortaleza (UNIFOR), Fortaleza, Brazil, **7** Civil and Environmental Engineering Department, University of California Berkeley, California, Berkeley, United States of America

* hmakse@ccny.cuny.edu

**Data Availability Statement:** Source codes of all algorithms used in this study are available at https://github.com/makselab/COVID19 and https://

## Abstract

The spread of COVID-19 caused by the SARS-CoV-2 virus has become a worldwide problem with devastating consequences. Here, we implement a comprehensive contact tracing and network analysis to find an optimized quarantine protocol to dismantle the chain of transmission of coronavirus with minimal disruptions to society. We track billions of anonymized GPS human mobility datapoints to monitor the evolution of the contact network of disease transmission before and after mass quarantines. As a consequence of the lockdowns, people's mobility decreases by 53%, which results in a drastic disintegration of the transmission network by 90%. However, this disintegration did not halt the spreading of the disease. Our analysis indicates that superspreading k-core structures persist in the transmission network to prolong the pandemic. Once the k-cores are identified, an optimized strategy to break the chain of transmission is to quarantine a minimal number of 'weak links' with high betweenness centrality connecting the large k-cores.

## Author summary

The emergence of the COVID-19 pandemic has revealed the importance of public measures to halt the spreading of highly infectious diseases. In this work, we implement a contact tracing network analysis over billions of anonymous GPS human mobility datapoints to monitor the evolution of the disease contact network. Our network analysis shows that a drastic reduction in people's mobility under mass lockdowns results in a drastic disintegration of the transmission network. However, this disintegration is not complete and the virus keeps spreading in the k-cores of the contact network. This result highlights the importance to perform digital contact tracing protocols in addition to the mass lockdowns to completely break the remaining transmission network. We then demonstrate an

github.com/makselab/COVID19_contact_detector or at the following link https://osf.io/x7g58/.

**Funding:** SDSR and JSA thank the Brazilian agencies CNPq [CNPq PQ Grant (Process 303765/2017-8) and CNPq Universal Grant (Process 436076/2018-7)], CAPES [CAPES PRINT Grant (Process 88887.311932/2018-00)] and FUNCAP [FUNCAP/CNPq PRONEX Grant (Process PR2-0101-00050.01.00/15)], and the National Institute of Science and Technology for Complex Systems [National Institute of Science and Technology for Complex Systems/ MCTI/CNPQ/CAPES/FAPS Grant (Process 465618/2014-6)] for financial support. MS acknowledges support from SoBigData++ (EC grant number 871042). This work was partially supported by NIBIB and NIMH through the NIH BRAIN Initiative Grant R01 EB028157. The funders had no role in study design, data collection and analysis, decision to publish, or preparation of the manuscript.

**Competing interests:** The authors have declared that no competing interests exist.

optimized strategy to break the remaining transmission chain, which first identifies the k-cores and then quarantines a minimum number of 'weak links' with high betweenness centrality connecting the large k-cores.

## Introduction

In the absence of vaccine or treatment for COVID-19, state-sponsored lockdowns have been implemented worldwide to halt the spread of the ongoing pandemic creating large social and economic disruptions [1–4]. In addition, some countries have also implemented digital contact tracing protocols to track the contacts of infected people and reinforce quarantines by targeting those at high risk of becoming infected [5–14]. Here we develop, calibrate, and deploy a contact tracing algorithm to track the chain of disease transmission across society using big-data from mobile phone geolocalization. We then search for intelligent quarantine protocols to halt the epidemic spreading with minimal social disruptions [15–20].

Mobile phones or similar devices provide digital sources of information on human mobility and therefore offer a promising way to automate outbreak location detection. Mobile datasets generally consist of an ID associated with each user, a timestamp of the user location, and a location provided as latitude/longitude, which places the user in space. In [21] the authors propose a method to identify outbreak locations of point-source outbreaks from geo-located GPS movement data of affected individuals as recorded from mobile phones. In [22] the authors investigate whether the observed discrepancies between mobile phone datasets affect the results of epidemic simulations. Ferretti *et al.* [13] showed that a contact tracing App can achieve epidemic control if used by enough people without resorting to mass quarantines. Other works combined cross-sectional survey and GPS data. For example, in [23] the authors define a contact tracing strategy that is likely to identify a sufficient proportion of infected individuals such that subsequent spread could be prevented. The solutions proposed often rely on using GPS data alone or combining GPS with self-reported infections (through a mobile app or questionnaire). Our study uses two complementary datasets. The first includes data from 'Grandata-United Nations Development Programme partnership to combat COVID-19 with data' [24]. It is composed of anonymized global positioning system (GPS) data from a compilation of hundreds of mobile applications (apps) across Latin America that allow to track the trajectories of people (users). The data identify each mobile phone device with a unique encrypted mobile ID and specifies its latitude and longitude location through time, which is encoded by a geohash with 12 digits precision. Typically, this dataset generates $\approx 450$ million data points of GPS location per day across Latin America (S1 Appendix, section 1). Our analysis is focused on the state of Ceará, Brazil, where we track the geolocation of over a quarter million unique users generating over half a billion GPS datapoints during the three months period of our study. The second dataset is an anonymized list of confirmed COVID-19 patients obtained from the Health Department authorities from the City of Fortaleza, Ceará, Brazil. The dataset contains the geohash of the residential address, the SARS-CoV-2 test detection date, and the first day of symptoms for each patient infected with COVID-19 in the city of Fortaleza over the studied period, which starts with patient zero arriving in the city and being detected on March 8, 2020. This dataset is used with the consent of the local health authorities in Fortaleza, Ceará and constrained the possibility of retrieving the chain of transmission of the virus to the state of Ceará. We cross-match the location of the residential address of each patient with the GPS geolocation from the mobile phone dataset, thus obtaining the encrypted mobile ID of the patients (S1 Appendix, section 7). We then trace the

geolocalized trajectories of COVID-19 patients during a period -14/+7 days from the onset of symptoms to look for contacts of the infected person. These contacts define the chain of transmission of the disease which is obtained using the model described below.

## Results

### COVID-19 model

The COVID-19 spreading model is represented by a Susceptible-Exposed-Infectious-Recovered (SEIR) process [16] and considers the epidemiological profile depicted in Fig 1A. An epidemiological profile is characterized by the incubation time (time from exposure, E, to onset of symptoms), latency (from exposure to onset of infectiousness, I), infectious period (the period over which the patient is contagious), and the extend of the disease (from the onset of symptoms until recovery or death, R). The values of the corresponding times of SARS-CoV-2 depicted in Fig 1A are obtained from the literature [25–27]. A crucial feature to notice from the epidemiological profile is that the onset of the infectiousness period occurs before the onset of symptoms. In other words, the latency is shorter than the incubation period, and the patient becomes contagious before she/he starts to feel the symptoms of the disease. Furthermore, the peak of infectiousness, that is, when the patient is at its most contagious stage, occurs, in average, about a day or two before the onset of symptoms, according to the study done in [25] at the beginning of the pandemic. These numbers are crucial to understand the rapid spread of COVID-19. They imply that when the patient starts with the symptoms of the disease, she/he has already transmitted the virus to the majority of its infected people during

## Epidemiological profiles

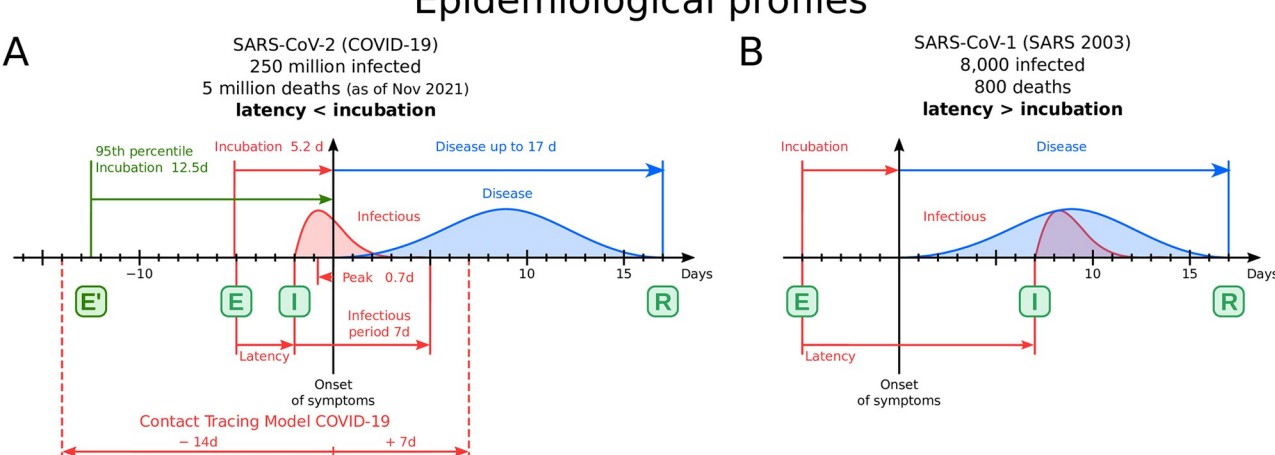

**Fig 1. Infectiousness profiles of SARS-CoV-1 and SARS-CoV-2. (A)** Infectiousness profile of coronavirus SARS-CoV-2 responsible for COVID-19. The COVID-19 pandemic is modeled by a SEIR model. From exposure (E) the virus is incubated in average for 5.2 days (12.5 days 95th percentile), starting the symptoms 2 days after infectiousness (I) and lasting the disease up to 17 days to recover (R). We use a window -14/+7 days from the first symptoms to detect infectious and exposure. **(B)** Infectiousness profile of coronavirus SARS-CoV-1 responsible for SARS-2003. Data obtained from [25]. As opposed to COVID-19, we note that in this case the latency is longer than the incubation period, and the peak of infectiousness then appears after the onset of symptoms. Thus, when the patients present its first symptoms, upon isolation, the transmission of disease is interrupted. In this case, isolating the patients after the symptoms is an effective way to control the pandemic. On the contrary, COVID-19 in (A) is characterized by a latency shorter than incubation, and, even more troublesome, with a pre-symptomatic peak of transmission appearing before the onset of symptoms. Thus, in this case, even if the patient isolates after the symptoms appear, most of its infections have occurred already. This indicates that the only way to stop the chain of transmission of COVID-19 is by going into the past, before symptoms, and performing contact tracing to capture and isolate the contacts of the infected person before the symptoms have appeared. This crucial difference in the epidemiological profiles of these two coronaviruses might explain why SARS was contained successfully in 2003 producing around 8,000 infections and 800 deaths, while COVID-19 kept spreading reaching a much larger worldwide population of 250 million infections and 5 million deaths as of November 2021.

the previous two days. Therefore, even if the patient isolates him/herself after feeling sick, the main transmissions have already occurred.

This peculiar feature of the new coronavirus implies that the only way to stop the chain of transmission (in the absence of vaccines) is to perform contact tracing to track the past contagious contacts of the patient and isolate them. That is, we need to go back in time to identify the contacts that have already occurred before the patient reports the symptoms to the health authorities. Without contact tracing, the chain of transmission cannot be broken, even if the patient enters into isolation after the onset of symptoms. This situation is exacerbated due to the existence of asymptomatic cases, i.e., infected people who do not feel symptoms and can potentially transmit the disease without knowing it.

The relation between latency and incubation of SARS-CoV-2 is inverted in coronavirus SARS-CoV-1 responsible for the SARS pandemic in 2003. As we see in the epidemiological profile of SARS-CoV-1 in Fig 1B, in this case, the latency is longer than the incubation period, according to studies reported in [25]. Patients of SARS 2003 become contagious a few days after the appearance of symptoms. In this case, upon isolation of the patient after reporting symptoms, the chain of transmission can be successfully broken. Thus there is no need to perform contact tracing back in time before the onset of symptoms since all contagious contacts happen during the manifestation of the disease. This situation could explain why SARS in 2003 was contained successfully without spreading worldwide, as this coronavirus infected "only" about 8,000 people with around 800 deaths worldwide. On the other hand, the new coronavirus spread to all continents infecting 250 million people and causing 5 million deaths as of November 2021 across the world. Furthermore, countries that implemented effective earlier contact tracing protocols, like South Korea and China [5–12], were able to contain the pandemic more successfully than counties who did not implement contact tracing protocols. Inspired by this evidence, our study is an attempt to scientifically show how contact tracing works in a real setting.

The infectiousness period of an infected person starts 2 days before and lasts up to 5 days after the onset of symptoms [25]. In this paper, we added two extra days to be conservative in capturing the contacts since the number of days comes from statistical estimations of the different periods characterizing the epidemiological profile of the disease, see Ref. [25]. Thus, in principle, to trace those people potentially infected by COVID-19 patients, we track contacts 4 days before and 7 days after the reported date of first symptoms (see Fig 1A). In addition, we extend the tracing period further back in time to also consider exposures that could come from asymptomatic cases (S1 Appendix, section 7). Exposures start the incubation period of the infected person which can occur up to 12.5 days before onset of symptoms (5.2 days on average, 95% percentile 12.5 days [26, 27], Fig 1A). To conservatively trace these exposure events, we add $\sim 2$ days to this incubation period and obtain the widely used 14 days period. Hence, to trace transmission and exposure cases, we perform contact tracing over -14/+7 days from onset of symptoms (Fig 1A). As noted above, the peak of infectiousness as well as 44% (95% confidence interval, 25–69%) of infected cases occur during the pre-symptomatic stage [25]. Thus, performing contact tracing is essential to stop the spreading of the disease.

## Contact model

The GPS geolocation of the trajectories of both infected and susceptible people is used to trace several layers of contacts in the transmission network using the following model (S1 Appendix, section 2). A contact at time stamp $n$ is initiated with an infected user (source) at time $t_0$ (see Fig 2A). The timestamp $n$ enumerates each GPS datapoint, while $t_n$ refers to the actual time attached with that point. At $t_0$ we draw a contact area as a circle centered in the source position

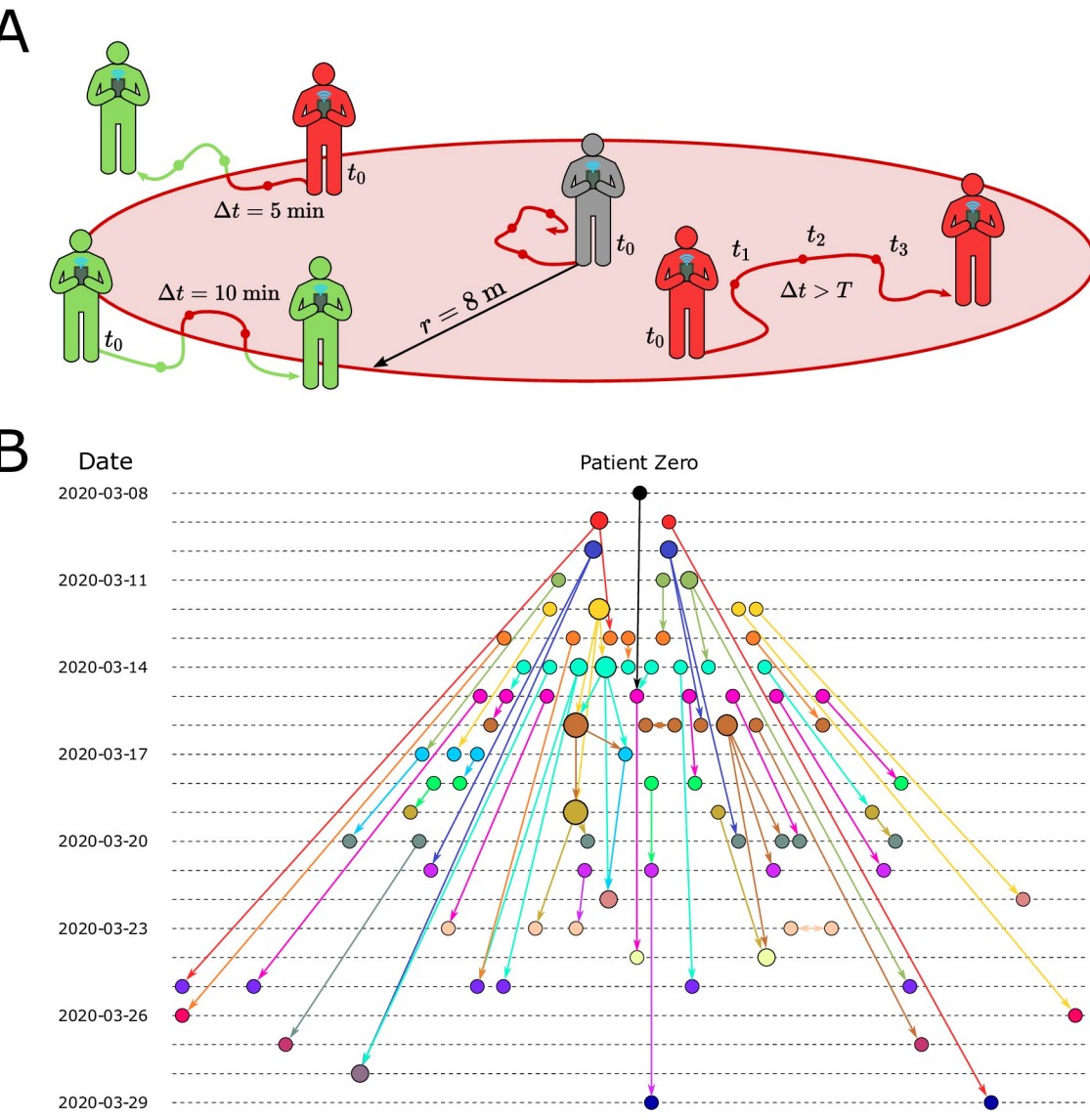

**Fig 2. COVID-19 contact model.** (A) Contact area used in the contact tracing model. The grey person is at the first datapoint of the source at $t_0$. We collect all datapoints for every user in a $T = 30$ min forward window ($t_1, t_2, t_3, \ldots, t_0 + T$) within an 8 m circle from the initial position. For each target (green and red) we compute the average position and the time spent inside the contact area (red part of the trajectory line). (B) Partial transmission tree of outbreak of confirmed SARS-CoV-2 infection identified by contact tracing during calibration in the month of March 2020. Links goes from the source of infection to the target. The colors represent the day of first symptoms for each node and size is the out-degree.

with a radius $r$. We then gather all the GPS datapoints from susceptible users (targets) that enter the contact area from $t_0$ to $t_0 + T$, where $T$ is the total exposure time. We follow the trajectories of source and target within the time-space area and compute the probability of infection at time stamp $n$ as $p_i[n] = p_d[n] \cdot p_t[n]$, where $p_d[n]$ is the spatial component, and $p_t[n]$ is the temporal component. When the average overlap between source and target is zero, then $p_d[n] = 1$, and when the overlap is $2r$, then $p_d[n] = 0$. On the other hand, when the exposure time $\geq T$, then $p_t[n] = 1$, and decreases to $p_t[n] = 0$ as the exposure time decreases (S1 Appendix, section 2). The probability $p_d[n]$ quantifies the contact probability for two users in the

same area defined by $r$. A contact requires non only a space overlapping but also a time overlap, $p_t[n]$, which quantifies the probability that two users met based on the time commonly spent in the same area. We then combine these two probabilities for each timestamp $n$ into their product.

Contacts with low probability of infection $p_i[n]$, but repeated throughout time, can also infect the target. To incorporate this effect in the model, we define the probability of infection for a series of repeated contacts $P_i[n]$ as a recursive formula from time 1 to $n$ with $P_i[0] = 0$:

$$P_i[n] = p_i[n](1 - P_i[n-1]) + P_i[n-1]. \tag{1}$$

The iteration of contacts between source and target, $P_i[n]$, generates higher probability of infection than a single contact $p_i[n]$. This means that there is a difference between a short single contact between two people and short repeated contacts between the same people. The latter scenario should have a larger probability than the former to become infected. While the distribution of $p_i[n]$ is homogeneous without a clear threshold for an infectious contact, $P_i[n]$ presents a very polarized distribution where the values are accumulated in the extremes: $P_i = 0$ or $P_i = 1$ (see S1 Fig). Thus, $P_i[n]$ is better indicator than $p_i[n]$ to separate infectious from non-infectious contacts. A contact is then considered infectious when this probability exceeds a certain threshold, $P_i[n] > p_c$. The hyperparameters of the contact model ($T$, $r$, $p_c$) are obtained by calibrating the model using only the contacts between infected people to reproduce the basic reproduction number $R_0 = 2.78$ in Ceará in the month of March, 2020 (S1 Appendix, section 3). We obtain $T = 30$ min, $r = 8$ m and $p_c = 0.9$. Thus, a contact is defined with probability one when exposure is at least 30 minutes within a distance $\ll 8$m. This calibration procedure provides the partial transmission tree of the outbreak from patient zero to the end of the calibration period shown in Fig 2B.

## Transmission network model

Next, we create the contact network of coronavirus transmission by first tracing the trajectories of confirmed COVID-19 patients to search for contacts -14/+7 days from the onset of symptoms using the above model. From the first contact layer, we add four layers of contacts to constitute the contact network of transmission that is used to monitor the progression of the pandemic. The time-varying network is aggregated to a snapshot defined over a time window of a week [16] (S1 Appendix, section 7). We find that other aggregation windows give similar results as presented.

Next, we analyze the spatio-temporal properties of the contact network. The government of the State of Ceará imposed a mass quarantine on March 19, 2020 which led to a decrease in people's mobility by 56.5% as shown in Fig 3A. During the lockdown, only the displacements of essential workers were allowed. A large decrease in mobility is also observed across all Latin America, see [24].

## Giant connected component (GCC)

To understand the effect of the lockdown on the contact network, we think by analogy with a "bond percolation" process [16, 17, 28]. In bond percolation, the network connectivity is reduced by removing a small fraction of links (bonds) between nodes, and the global disruption in network connectivity is monitored by studying the normalized size of the giant connected component (S1 Appendix, section 4). Following this analogy, the lockdown acts as a percolation process, and therefore we monitor the GCC of the transmission network before and after the lockdown. We find a large decrease in the size of the GCC [16, 28] within 6 days

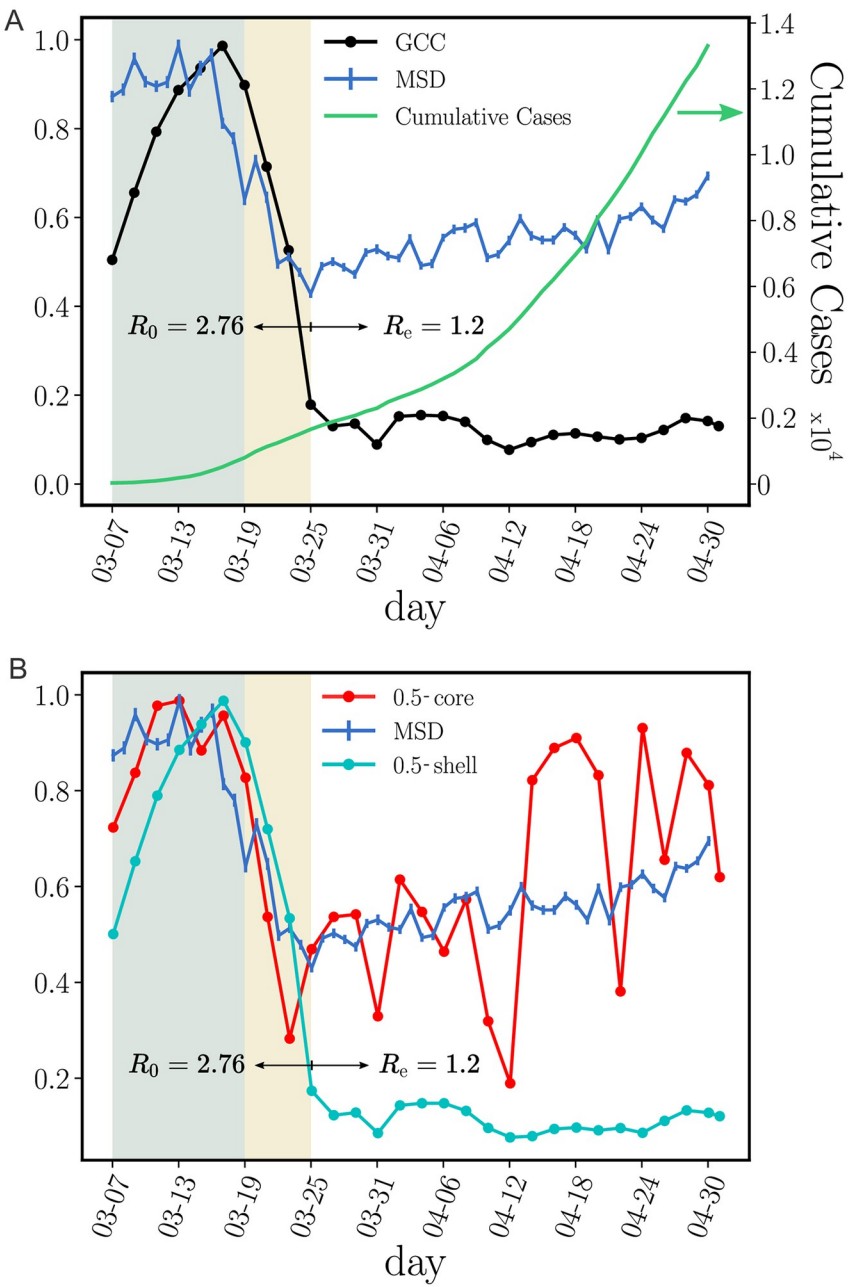

**Fig 3. Structural components of transmission networks across the lockdown. (A)** Evolution for different metrics in Ceará, Brazil, previous to the mass quarantine (grey area), right after the imposed quarantine (yellow area) and later. The plot shows the root mean square displacement (MSD) normalized by the maximum value over the total period (blue), the cumulative number of cases (green) and the size of the GCC normalized by the maximum value over the total period (black). The uncertainty corresponds to the standard error (SE). The mobility data is showcased in the Grandata-United Nations Development Programme map shown in https://covid.grandata.com. The initial rise in GCC is due to the lack of data before March 1. **(B)** The plot shows the 0.5-core size (red), the 0.5-shell size (cyan) all normalized by their respective maximum value pre-lockdown. While the size of the 0.5-shell is reduced drastically during the lockdown, the 0.5-core was not reduced as much and keeps increasing, contributing to sustain the pandemic. The 0.5-core seems to follow the trend in the MSD, which we plot again to show this trend.

of the implementation of the lockdown on March 19, when the GCC is almost fully dismantled decreasing by 89.6% of its pre-lockdown size (Fig 3A).

Despite the disintegration of the GCC, the cumulative number of cases kept growing albeit at a lower rate (Fig 3A). We find that the mass quarantine was able to reduce the basic reproduction number from $R_0 = 2.78$ before lockdown to an effective reproduction number of $R_e = 1.2$ after the lockdown (Fig 3A). Despite this disruption in the network connectivity, $R_e$ has not decreased below one, as it would have been needed to curb the spread of the disease.

The drastic reduction in the GCC is visually apparent in the contact networks in Fig 4. Before lockdown on March 19 (Fig 4A), the network is a strongly-connected unstructured "hairball". Eight days into the lockdown on March 27 (Fig 4B), the network has been untangled into a set of strongly-connected modules integrated by tenuous paths of contacts. This structure is even more pronounced a few weeks later on April 28 (Fig 4C).

## Superspreading k-core structures

The highly connected modules found in Fig 4B and 4C are k-core structures [30–33] of higher complexity than the GCC (which is a 1-core), that are known to sustain an outbreak even when the GCC has been disintegrated [16, 33]. The k-core of a graph is the maximal subgraph in which all nodes have a degree (number of connections) larger or equal than $k$ [30–33]. The k-shell is the periphery of the k-core and is composed by all the nodes that belong to the k-core but not to the (k+1)-core (S1 Appendix, section 5). The k-core is obtained by iteratively pruning the nodes with degree smaller than $k$. For instance, the 3-core is obtained by removing the 1-shell and 2-shell in a k-shell decomposition process (S1 Appendix, section 5). Thus, all nodes in a k-core have at least degree $k$, and are connected to other nodes with degree at least $k$ too. K-cores are nested and can be made of disconnected components (see S4 Fig). High k-cores are those with large $k$ up to a maximal $k_{core}^{max}$, and constitute the inner most important part of the network. In theory, the high k-cores are known from network science studies to be the reservoir of disease transmission persistence [16, 33]. On the contrary, low peripheral k-shells (see S2 Fig) do not contribute as much to the spread as the high inner k-cores.

Fig 3B shows that despite the disappearance of the GCC, there is a significant maximal k-core that was not dismantled by the mass quarantine. The figure shows that the outer k-shells of the transmission network (i.e., the 0.5-shell defined as the union of the k-shells with $k = 1, 2, ..., \lceil 1/2\, k_{core}^{max} \rceil - 1$) are disintegrated in the lockdown, decreasing by 91% with respect to their pre-quarantine size, in tandem with the GCC. However, the inner k-core (i.e., the 0.5-core defined as the k-core with $k = \lceil 1/2\, k_{core}^{max} \rceil$) persists in the lockdown. The figure shows that the decrease of the 0.5-core is only 50% compared to the 91% decrease of the 0.5-shell; the former even increases slightly at the end of April, following the same trend in mobility (see Fig 3B). This process is visually corroborated in the evolution of the networks seen from Fig 4A and 4C where we observe the disappearance of the peripheral k-shells and the persistence of the maximal k-core. Indeed, the unessential contacts in the peripheral k-shells may have been first pruned during social distancing.

Using numerical simulations, we corroborate previous results indicating that the infection can persist in these high k-cores of the network while virus persistence in outer k-shells is less important [16, 33]. We use a SIR model on the transmission network (see S15 Fig) showing that the maximal k-cores of the network sustain the spreading of the disease more efficiently than the outer k-shells. Thus, the maximal k-core components of the contact network are plausible drivers of disease transmission. Apart from this structural explanation (i.e., k-core), epidemiological factors may also play a role in the persistence of the disease, such as a transition

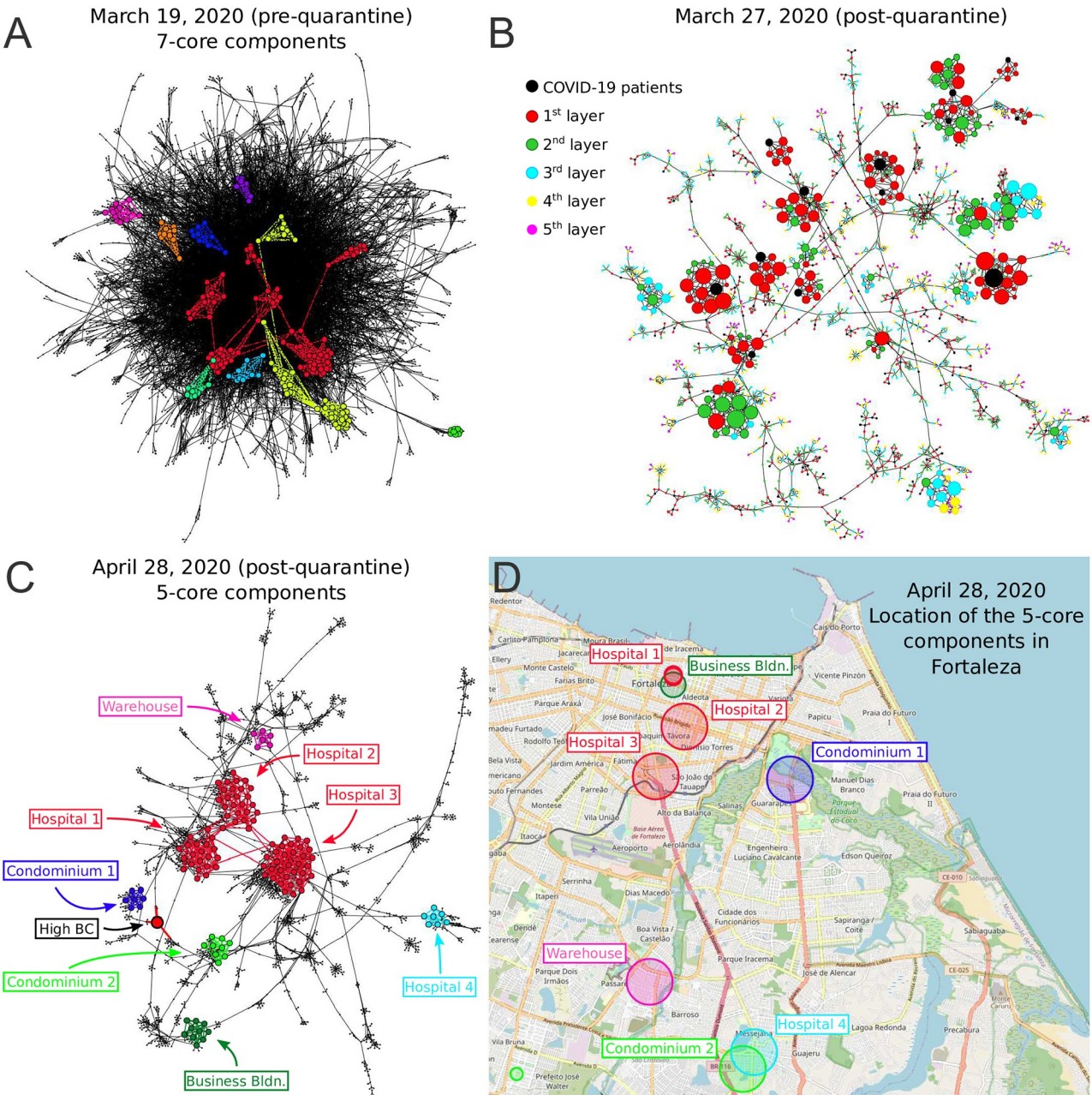

**Fig 4. Evolution of GCC and k-cores over the quarantine.** Disease transmission networks in the state of Ceará over time before and after the lockdown on March 19, 2020. **(A)** Transmission network on March 19 (pre-lockdown). A hairball highly-connected network is observed. The disconnected components of the 7-core ($k_{core}^{max} = 12$ in this network) are colored. These components are well connected into the hairball network as expected since mobility and connectivity is high. **(B)** The pre-quarantine hairball in **(A)** has been untangled and the k-cores have emerged 8 days into the lockdown on March 27. Here, we color the nodes according to layers of the transmission network starting at COVID-19 patient (black nodes). Size of nodes is according degree. **(C)** Network on April 28 including the components of the 5-core in different colors ($k_{core}^{max} = 7$ for this network). Visible is the high betweenness centrality node representing the weak-link of this k-core. **(D)** We plot the location of the contacts in the map of Fortaleza constituting the components of the 5-core of the April 28 in **(C)**. The size of the circles in the map corresponds to the number of contacts inside each location. The colors correspond to the clusters of the 5-core in **(C)**. The 5-core sustaining transmission is composed of clusters of contacts localized in hospitals, large warehouses and business buildings. Hospital 3, one of the largest in Fortaleza, constitutes the maximal $k_{core}^{max} = 7$ of the pandemic. The underlying map comes from the Folium library of Python: https://github.com/python-visualization/folium which relies on the OpenStreetMap project [29].

of the disease to vulnerable communities with high demographic density, or with large inhabitants per household where isolation is poorly fulfilled.

When we plot the geolocation of the contacts forming the maximal k-core in the map of Ceará, we find that these contacts take place in highly transited areas of the capital Fortaleza, such as hospitals, business buildings, warehouses as well as large condominiums, see Fig 4D. These contacts generate superspreading k-core events that generalize the conventional notion of superspreaders, which refer mainly to individuals with large number of transmission contacts [34–36]. However, connections are not everything [18, 19]. K-core superspreaders not only generate a large number of transmission contacts, but their contacts are also highly connected people, and so forth.

## Optimized quarantine

The existence of k-cores in the transmission network suggests that a more structured quarantine could be deployed to either isolate or destroy those cores that help maintain the spread of the virus. We perform an optimal percolation analysis [18–20] to find the minimal number of people necessary to quarantine that will dismantle the transmission network. We compare different strategies to find the best among them to break the network by ranking the nodes based on (1) the number of contacts (hub-removal) [16, 18, 19], (2) the largest k-shells and then by the degree inside the k-shells [16, 33], (3) the collective influence algorithm for optimal percolation [20], (4) the generalized k-core strategy [37], and (5) betweenness centrality [38–41].

Fig 5B shows the normalized size of the GCC versus the fraction of removal nodes following different strategies, as well as a random null model of removal in a typical network under lockdown in April 28 (March 19 pre-lockdown results are plotted in S15 Fig). While the disease can persist in the k-cores (Fig 5A), quarantining people directly inside the maximal k-core is not an optimal strategy. The reason is that k-cores are populated by hyper-connected hubs that require many removals to break the GCC [40] (around 7%, see Fig 5B). For the same reason, removing directly the hubs is not the optimal strategy either, since the hubs are within the maximal k-core and not outside. A collective influence strategy [20] improves over hub-removal since it takes into account how hubs are spatially distributed, yet, it is far from optimal. A generalized k-core strategy, which consists in sequentially removing the nodes in the k-leaf (where $k = k_{max}$), has been recently reported to be more suitable to study spreading behavior [37]. Fig 5B shows that, in this case, it performs similarly like k-core. The reason for this can be found in the tree structure of the network and its low average degree. Clearly, Fig 5B shows that the best strategy is to quarantine people by their betweenness centrality. By removing just the top 1.6–2% of the high betweenness centrality people, the GCC is disintegrated. This result is consistent with the particular structure of the transmission networks seen in Figs 4B and 4C and 5.

The betweenness centrality of a node is proportional to the number of shortest paths in the network going through that node. Thus, given the particular structure of the networks in Figs 4B and 4C and 5C, the high betweenness centrality nodes are the bottlenecks of the network, i.e., loosely-connected bridges between the largely-connected k-cores components. These connectors are the "weak links", fundamental concept in sociology proposed by Granovetter [42], according to which, strong ties (i.e., contacts in the k-cores) clump together forming clusters. A strategically located weak tie between these densely "knit clumps", then becomes the crucial bridge that transmits the disease (or information [42]) between k-cores. These weak links are people traveling among the different k-cores components allowing the disease to escape the cores into the rest of society. These bridges are displayed in the network of Fig 5C as the yellow, blue and red nodes. The removal of these high betweenness centrality people disconnects

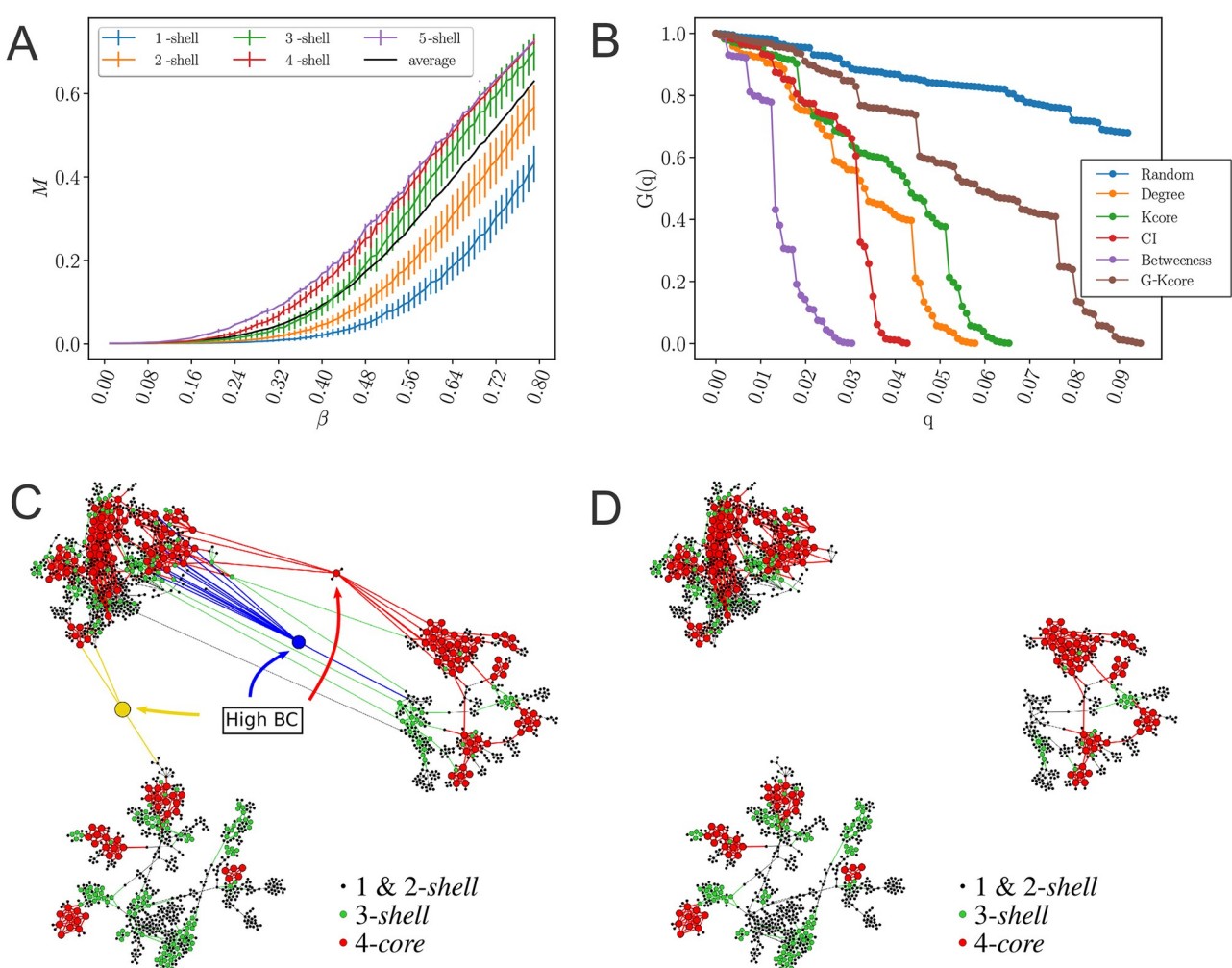

**Fig 5. Weak links and k-cores. (A)** Average size of infected population, $M$ [33], in an outbreak average over all starting nodes in a k-shell as a function of the probability of infection $\beta$ for a SIR model on the network in Fig 4C during the lockdown. The black is the average value over all the network. The average divides the k-shell contribution to the spreading of the virus in two groups: above and below the average. The 0.5-cores have maximal spreading and the 0.5-shell have minimal spreading. Error bars correspond to a confidence interval of 95%. **(B)** Optimal percolation analysis performed over the network in Fig 4C during the lockdown in following different attack strategies and their effect on the size of the largest connected component $G(q)$ versus the removal node fraction, $q$. Nodes are removed (in order of increasing efficiency): randomly (blue); by the highest k-shell followed by high degree inside the k-shell [33]; by highest degree (orange); by collective influence (red) [20]; by the highest generalized k-core (brown) [37]; and by the highest value of betweenness centrality (green) [38, 39]. After each removal we re-compute all metrics. The most optimal strategy among those studied is removing the nodes by the highest value of betweenness centrality. **(C)-(D)** Effect of removing three high betweenness centrality nodes shown in Fig 5B in the network of Fig 4C. **(C)** We show the 2-core component of the network after the removal of 12 high betweenness centrality nodes. The red node is the one with the highest betweenness centrality value (next node to remove, 13th) and the blue node is the 14th removal. Different k-cores and k-shell are in different colors. **(D)** Network k-cores are disintegrated after the removal of the high BC nodes.

the k-core components of the network entirely, as shown in Fig 5D, halting the disease transmission from one core to the other [40, 43].

An important finding is that quarantining the large superspreading k-cores is neither optimal (as shown in Fig 5B, green curve) nor practical, since they are mainly comprised by chiefly essential workers who need to remain operational (Fig 4D). Thus, the best strategy, in conjunction with a mass quarantine, is then to disconnect these k-cores from the rest of the social network (Fig 5C and 5D), rather than quarantining the people inside the k-cores. This can be performed by quarantining the high betweenness centrality weak-links that simultaneously

preserve the operational k-cores. However, individuals belonging to the maximal k-cores should be tested at a higher frequency to promptly detect their infectiousness before the symptoms start, to help control the spreading inside the k-cores.

## Conclusion

Isolating the k-core structures by quarantining the high betweenness centrality weak links in the transmission network proves to be an effective way to dismantle the GCC of the disease while keeping essential k-cores working. While destroying the strong links and cores is a less manageable task to execute and control, isolating the weak links between cores is a more feasible task that will assure the dismantling of the GCC. In other words, if one core is infected, the disease will be controlled within that core and not extended to the rest of society.

It is worth stressing that the optimal strategy to break the transmission of the virus depends on the particular spreading dynamics of the disease, patterns of mobility, and strength of the quarantine applied to each region. As we show in Fig 4B, every centrality measure can, with a certain degree of disruption, dismantle the chain of transmission of the virus. As we can see from the same figure, the betweenness centrality provides the minimal number of nodes that need to be isolated to dismantle the chain of transmission as compared with the studied centralities. The reason why BC performs better than the other centralities can be found in the particular structure of the contact network left after the quarantine. As we show in Fig 4C a k-core structure appears due to the strict lockdown, during which only essential workers were allowed to go out. The lockdown essentially removes the majority of the links leaving only those inside the k-cores plus their weak links. These k-cores, which represent the virus reservoir, are generally located in hospitals, warehouses, and some particular condominiums since they are composed mainly of the essential workers who are allowed to circulate during the quarantine. The k-cores are connected by a few links, which work as bridges for the virus transmission. This particular network structure explains why a BC-based ranking is able to break the transmission chain with fewer removals than other centralities, since BC can identify better those bridges that connect the k-cores.

Thus, in the particular case of Fortaleza, we found that betweenness centrality provides the best ranking among the studied centralities to break the transmission chain. However, in another pandemic or even the same pandemic under a different quarantine protocol, the particular network structure that we found in Fortaleza may not appear. Therefore we do not expect that BC will always be the best method to break the transmission chain, and each particular case should be analyzed independently.

However, the strategy proposed here to use contact tracing and network theory is valid for any pandemic. This includes building and monitoring the GCC of transmission as a function of time by combing GPS data with patient-list data and then testing different centralities with the objective of finding the best strategy to break the GCC. Each pandemic and quarantine may lead to a different network structure with its concomitant optimal centrality. The proposed protocol is then to investigate all centralities as done in this study and find the strategy that would break the chain of transmission in the most optimal way.

As governments around the world have been trying to roll out digital contact tracing apps to curb the spread of coronavirus [5–12], our modeling suggests possible intelligent quarantine protocols that could become key in future phases of reopening economies across the world and, in particular, in developing countries where resources are scarce. Overall, our network-based optimized protocol is reproducible in any setting and could become an efficient solution to halt the progress of the COVID-19 pandemic worldwide drawing upon effective quarantines with minimal disruptions.

## Ethics statement

This study was approved by the Institutional Review Board (IRB) at City College of New York (Approval No: 2020–0423) and by the Comitê de Ética em Pesquisa at Universidade Federal do Ceará in accordance with Resolution CNS No 510. Patient data was used with the approval and consent from the Epidemiological Surveillance Department, Fortaleza Health Secretariat and the Mayor of the Prefeitura de Fortaleza, Ceará, Brazil.

## Supporting information

**S1 Appendix.** Detailed description of of the data acquisition and treatment (**Section 1**), of the theoretical tools mentioned in the main text (**Sections 2, 3, 4, 5 and 6**), and discussion on the extent to which the present results would hold under implementation in terms of robustness to data quality and coverage, sampling bias on demographics such as coverage of location, socio-economic status, age and gender and privacy (**Section 7**).
(PDF)

**S1 Fig. Transmission probability. (A)** Probability distribution of $p_i[n] = p_d[n] \cdot p_t[n]$ (orange) and the recursive form $P_i[n]$ defined in Eq (1) (blue). The $P_i[n]$ are polarized to 0 and 1 becoming the best thresholded metric to use to consider a contact as infectious. **(B)** Average value $< P_i[n] >_T$ as a function of the time window $T$ of the spatio-temporal contact area. $P_i[n]$ has a peak at $T = 30$ min; it decreases for $T > 30$ min and increase for $T < 30$ min as a function of $T$. The decreasing behaviour is what is expected, thus, 30 min is the minimum bound for the correct value of $T$.
(TIFF)

**S2 Fig. Network structure under k-shell decomposition. (A)** A sample network with 3 shells. The k-shell index $k_s$ is not necessarily associated with other centralities. Here, the hub of the network in black with $k = 7$ is in the 1-shell, $k_s = 1$. The two top node in betweenness centrality, highlighted in red, belong to the 2-shell and the 3-shell, respectively. The 1-core is equivalent to the GCC. **(B)** The nodes with $k_s = 1$ form the 1-shell, **(C)** the nodes with $k_s = 2$ form the 2-shell, and **(D)** the nodes with $k_s = 3$ form the 3-shell which is also the 3-core.
(TIFF)

**S3 Fig. K-cores of a network. (A)** We start the k-shell decomposition with a network configuration where every node has at least degree $k = 1$. This set of nodes forms a 1-core. **(B)** Then, every node with $k = 1$ is iteratively removed to obtain the 2-core. As one can see, the removal of these nodes changes the degree distribution. Thus, nodes are removed until all remaining nodes are left with $k \geq 2$. **(C)** Following the k-shell decomposition nodes are removed until we obtain the 3-core. The 3-core can be made of multiple disconnected clusters.
(TIFF)

**S4 Fig. K-cores decomposition.** Example of k-core and k-shell structure in the network plotted in Fig 3B obtained during the lockdown. Here the colors are set by the k-shell occupancy of each node. Each k-core is composed by the k-shell plus the (k+1)-core. The k-cores are nested structures. For instance, the 5-core in **(E)** is composed by the 5-shell (yellow nodes) and the 6-core, which, in turn, is composed by the 6-shell (in red) and the 7-core (in purple). Since the 7-core is the maximal k-core, $k_{\mathrm{core}}^{\max} = 7$ for this network, then the 7-core is also the 7-shell. In this network the 0.5-core is the 4-core and the 0.5-shell is composed by the 1-shell plus the 2-shell and the 3-shell. We notice how a given k-core can be composed of many disconnected components. For instance, the 6-core is composed by 5 disconnected components. This is important, since each component of a given k-core can be localized in different areas, like

different hospitals, in the map, see for instance, Fig 3C and 3D. It is also visually apparent that to destroy this network, a direct 'attack' to the high k-cores is not optimal. Instead, removing the high BC nodes that populate the lower k-shells is the best strategy. We plot each k-core in turn: **(A)** 1-core, **(B)** 2-core, **(C)** 3-core, **(D)** 4-core, **(E)** 5-core, **(F)** 6-core and **(G)** 7-core.
(TIFF)

**S5 Fig. Degree distribution of the contact network.** Degree distribution of the contact network before (blue) and after (orange) the quarantine.
(TIFF)

**S6 Fig. Evolution of the maximum k-core.** Evolution of maximum k-core index $k_{core}^{max}$ versus time previous to the quarantine (grey area), right after the quarantine (yellow area) and later. We see how the maximum k-core index drops drastically after the mass quarantine.
(TIFF)

**S7 Fig. Contact layers.** Contact layers or pre-symptomatic and asymptomatic captured by the model. Our treatment of asymptomatic cases is to increase the exposure period to -14 days to accounting for possible two-chains of infection as shown in the figure. Contacts between -2 days to -14 days from the day of first symptoms are more likely to be an exposure from an asymptomatic infected person. Contact from -2 days to +7 days from first symptoms are considered to be transmissions contacts from the patient.
(TIFF)

**S8 Fig. Sampling bias-coverage.** **(A)** Probability density function and **(B)** Cumulative distribution function of the fraction of the population per neighborhood in Fortaleza to the total population. We show the real distributions and the distributions from the apps GPS data. Both distributions pass a two-sample KS test indicating that we cannot reject the hypothesis that they come from the same distribution under the test.
(TIFF)

**S9 Fig. Sampling bias-HDI.** **(A)** Probability density function and **(B)** Cumulative distribution function of the fraction of the population per neighborhood with a given HDI in Fortaleza to the total population. We show the real distributions and the distributions from the apps GPS data. Two-sample KS test indicates that we cannot reject the hypothesis that the real and GPS sample come from the same distribution under the test, indicating lack of sampling bias under this test.
(TIFF)

**S10 Fig. Sampling bias-age.** **(A)** PDF and **(B)** CDF of age distribution in the GPS geolocalized data compared with the real patient data. We cannot reject the hypothesis that both samples come from the same distribution under KS statistical testing.
(TIFF)

**S11 Fig. Sampling bias-gender.** **(A)** PDF and **(B)** CDF of gender distribution in the GPS geolocalized data compared with the real patient data suggesting lack of bias.
(TIFF)

**S12 Fig. K-core persistence.** Persistence of people in the k-cores in the temporal networks. We plot the percentage of people in the cores from network to network. The persistance is calculated by the overlap of people in the k-shells from a time of observation to the next (three days later in this particular example).
(TIFF)

**S13 Fig. Robustness to false positive.** Normalized efficacy of BC centrality as a function of false positives in the report of infected people. A false positive is an individual who reported to have symptoms but was not infected with Covid-19. We plot the relative error in the determination of the minimal number of people to quarantine versus the false positive rate. The measure starts to deviate from linear behaviour beyond the error bars around 20% false positive rate. (TIFF)

**S14 Fig. GPS pings distribution.** Distribution of the time interval between GPS pings during all day and separated by day and night. (TIFF)

**S15 Fig. Weak links and k-cores pre-quarantine. (A)** Amount of infected population ($M = \sum \frac{M_i}{N}$ see [33]) when the spreading starts in a given node in a k-shell as a function of the probability of infection $\beta$ for a SIR model on the same network on March 19 in Fig 3A in pre-quarantine Ceará. The black is the average value over all the starting nodes in the network. The average divides the shell contribution to the spreading of the virus in two groups above and below the average. The 0.5-core composed of the 6-core ($k_{core}^{max} = 12$ in this network) which contains nodes from the 6-shell to the 12-shell, has maximal spreading. The 0.5-shell which is composed by the remaining shell from 1-shell to 5-shell has minimal spreading, below the average. **(B)** Optimal percolation analysis performed over the network in Fig 3A before the quarantine on March 19 in Ceará with different attack strategies and their effect on the size of the largest connected component $G(q)$ versus the removal node fraction, $q$. Depending on the strategy nodes are removed: randomly (blue), by the highest value of betweenness centrality (green) [38, 39], degree (orange), collective influence (red) [20], and by the highest k-shell followed by high degree inside the k-shell [33]. After each removal we re-compute all the metrics. The best strategy among those studied is removing the nodes directly by the highest value of betweenness centrality. (TIFF)

**S16 Fig. Size of the GCC over time.** The number of nodes (blue) and edges (oranges) in the GCC versus time. The initial increase in the number of nodes is artificial due to the fact that we perform contact tracing 14 days back for each patient and our data collection started in March 1. Thus the networks in the first two weeks have relatively lower contacts than the rest. (TIFF)

**S17 Fig. Size of the 0.5-core over time.** Evolution of maximum 0.5-core size versus time normalized by the size of the GCC. The proportion of these maximum k-cores keeps increasing after the quarantine. (TIFF)

## Acknowledgments

We are grateful to S. Alarcón-Díaz and I. Belausteguigoitia for discussions and J. Maciel, G. Sousa, J. A. P. Barreto, and R. Sousa from the Health Secretariat of Fortaleza for data curation of the patients dataset.

## Author Contributions

**Conceptualization:** Hernán A. Makse.

**Data curation:** Antonio S. Lima Neto, Matías Travizano.

**Formal analysis:** Matteo Serafino, Higor S. Monteiro, Shaojun Luo, Saulo D. S. Reis.

**Investigation:** Matteo Serafino, Higor S. Monteiro, Saulo D. S. Reis, Hernán A. Makse.

**Methodology:** Matteo Serafino, Higor S. Monteiro, Saulo D. S. Reis, Carles Igual, José S. Andrade, Jr, Hernán A. Makse.

**Visualization:** Matteo Serafino, Higor S. Monteiro, Saulo D. S. Reis, Hernán A. Makse.

**Writing – original draft:** Matteo Serafino, Higor S. Monteiro, Shaojun Luo, Saulo D. S. Reis, Carles Igual, Antonio S. Lima Neto, Matías Travizano, José S. Andrade, Jr, Hernán A. Makse.

**Writing – review & editing:** Matteo Serafino, Higor S. Monteiro, Shaojun Luo, Saulo D. S. Reis, Carles Igual, Antonio S. Lima Neto, Matías Travizano, José S. Andrade, Jr, Hernán A. Makse.

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
