## [Decision Letter · Decision Letter 0]

21 Sep 2021

Dear Prof. Makse,

Thank you very much for submitting your manuscript "Superspreading k-cores at the center of COVID-19 pandemic  persistence" for consideration at PLOS Computational Biology.

As with all papers reviewed by the journal, your manuscript was reviewed by members of the editorial board and by several independent reviewers. In light of the reviews (below this email), we would like to invite the resubmission of a significantly-revised version that takes into account the reviewers' comments.

We cannot make any decision about publication until we have seen the revised manuscript and your response to the reviewers' comments. Your revised manuscript is also likely to be sent to reviewers for further evaluation.

Sincerely,

Benjamin Muir Althouse

Associate Editor

PLOS Computational Biology

Thomas Leitner

Deputy Editor

PLOS Computational Biology

Reviewer's Responses to Questions

**Comments to the Authors:**

Reviewer #1: This paper implements a comprehensive contact tracing network analysis to find an optimized quarantine protocol to dismantle the chain of transmission of coronavirus with minimal disruptions to society. The authors track billions of anonymized GPS human mobility datapoints to monitor the evolution of the contact network of disease transmission before and after mass quarantines. The results reveal here are timely and interesting. However, there are some issues that need to be clarified.

Two data sets are chosen to be studied in the paper. One is the Grandata united nations development programme partnership to combat covid and the the other is an anonymized list of confirmed covid patients obtained by the health department authorities from the two countries. There are some mobility data available covering a wide range of regions out there. The authors may want to comment on the choice of the datasets and limit their conclusion about the findings.

The infectiousness period of an infected person starts two days before and lasts up to five days after the onset of symptoms. However, this is not directly related to the problem and data studied in this paper. The decision to adding two days tothe limits should be commented.

Both cc and k core have been studied in this paper. However, it is recently reportly that generalized k core is more suitable for the study of spreading behavior. Therefore, this aspect should be discussed and compared thoroughly.

Another concern is regarding the optimal quarantine. The inconsistencies between different centralities should be explained better. Do you expect the same qualitative results for other similar infectious disease or covid related data in other capabilities? In other words, how general are the obtain results? The resilience of the method should not be overlooked.

Reviewer #2: The presented manuscript proposed an interesting contention strategy for the spreading phenomena in contact networks obtained from real GPS human mobility data. The authors create a contact tracing network that presumes to have the full contagion network. Then, they conclude k-core structures persist in the transmission network even when an extreme measure such as a lockdown is applied maintaining the spreading activity. This suggests that an optimized isolation measure can be found to avoid contagion. After trying different centrality measures the authors found that the betweenness centrality is the best breaking the transmission network. I found this preprint very interesting, well-written, and easy to follow. Furthermore, it contains enough novel results to be published in PLOS Computational Biology.

Reviewer #3: # Summary

The manuscript conducts a contact tracing network analysis matching GPS and confirmed cases data applied to the case of COVID-19. They propose an effective way to break the network of transmission based on data from the state of Ceará, in Brazil. This GPS data allow monitoring the mobility of users and build a contact network identifying temporal changes before and after the lockdown and the persistence of $k$-cores, linked to super-spreading events. The main finding is that it is possible to break the transmission tree quarantining those with high betweenness centrality, linking the maximum $k$-cores with the rest of the population. The work is original and of high importance, with rigorous network and statistical analyses, very well documented in the Supplementary Material.

The authors state that the research followed ethical guidelines to treat personal data, not allowing to identify anyone, and has the approval of the Epidemiological Surveillance Department of Ceará.

# Some points

- I had some questions about the sampling bias and the fact that only a few confirmed cases were linked to the GPS data. I suggest empathizing that these analyses were also performed and cite the Supplementary Material more often in the main text.

- line 18: typo -> "teh state"

- line 20: "both states"? Only Ceará is mentioned before. As I could see in the SM, there is a "Puebla" state that was not used. By the way, in the caption of Table S1 "Puebla" is also mentioned.

- line 40: I suggest changing $\\sim$ by $\\approx$ if the authors agree.

- line 49: what is the difference between timestamp $n$ and time $t$?

- lines 151-158: in Ref. DOI:10.1103/PhysRevE.98.012310 the authors show that the maximum $k$-core can be the driver of disease transmission in contact networks. I was wondering if it has relation with the case presented here.

- How is the degree distribution $P(k)$ of the network? Does it keep the shape over time, taking snapshots aggregating the network over different time windows before and after the lockdown?

- page 25 of SM: typo? -> "These GSP-based apps"

- Notation: The notation about $k$-cores and $k$-shells can be improved. Sometimes they appear as 1-shell (line 128) or 1 $k$-shell (Fig. 4a), while 0.5-kcore is also used. I would suggest to use 1-shell and 0.5-core, instead, to match the $k$-shell and $k$-core pattern, respectively.

**Have the authors made all data and (if applicable) computational code underlying the findings in their manuscript fully available?**

Reviewer #1: Yes

Reviewer #2: Yes

Reviewer #3: Yes

PLOS authors have the option to publish the peer review history of their article (what does this mean?). If published, this will include your full peer review and any attached files.

Reviewer #1: No

Reviewer #2: No

Reviewer #3: No
---

## [Decision Letter · Decision Letter 1]

25 Jan 2022

Dear Prof. Makse,

We are pleased to inform you that your manuscript 'Digital contact tracing and network theory to stop the spread of COVID-19 using big-data on human mobility geolocalization' has been provisionally accepted for publication in PLOS Computational Biology.

Best regards,

Benjamin Althouse

Associate Editor

PLOS Computational Biology

Thomas Leitner

Deputy Editor

PLOS Computational Biology

Reviewer's Responses to Questions

**Comments to the Authors:**

Reviewer #1: I have no further comment on this paper.

Reviewer #2: The authors have addressed the other reviewers concerns, I think the manuscript deserves to be published.

Reviewer #3: The authors revised the whole manuscript and I am satisfied with the answers to the questions raised in the review process.

**Have the authors made all data and (if applicable) computational code underlying the findings in their manuscript fully available?**

Reviewer #1: Yes

Reviewer #2: Yes

Reviewer #3: Yes

PLOS authors have the option to publish the peer review history of their article (what does this mean?). If published, this will include your full peer review and any attached files.

Reviewer #1: No

Reviewer #2: No

Reviewer #3: No

---

## [Editor Report · Acceptance letter]

1 Apr 2022

PCOMPBIOL-D-21-00586R1 

Digital contact tracing and network theory to stop the spread of COVID-19 using big-data on human mobility geolocalization

Dear Dr Makse,

I am pleased to inform you that your manuscript has been formally accepted for publication in PLOS Computational Biology. Your manuscript is now with our production department and you will be notified of the publication date in due course.

With kind regards,

Olena Szabo
